Soil polluted system shapes endophytic fungi communities associated with Arundo donax: a field experiment

Wang Xiaohui
Wang Yao
Sun Yingqiang
Wang Keyi
Yang Junbo
Zeng Danjuan
Mo Ling
Liao Jianxiong
Peng Qianshu
Yao Yu
Pu Gaozhong pukouchy@163.com
Guangxi Key Laboratory of Plant Conservation and Restoration Ecology in Karst Terrain, Guangxi Institute of Botany, Guangxi Zhuang Autonomous Region and Chinese Academy of Sciences , Guilin , China
Banaszak Anastazia
Electronic publication date: 2025 Jan 10
Publication date: 2025
Volume: 13
Electronic Location ID: e18789
Received 2024 Jul 4; Accepted 2024 Dec 10
Copyright: © 2025 Wang et al.
Copyright year: 2025
Copyright holder: Wang et al.
License: This is an open access article distributed under the terms of the Creative Commons Attribution License, which permits unrestricted use, distribution, reproduction and adaptation in any medium and for any purpose provided that it is properly attributed. For attribution, the original author(s), title, publication source (PeerJ) and either DOI or URL of the article must be cited.
License URL: https://creativecommons.org/licenses/by/4.0/

Keywords: Contaminated soil, Red mud soil, Phytoremediation, High throughput sequencing, Microbial network

Funding: National Natural Science Foundation of China 32460312 Basic research fund of Guangxi Academy of Sciences CQZ-D-1904 Fundamental Research Fund of the Guangxi Institute of Botany, China 24010 Key R&D Program of Guangxi, China AB22035038 Light of West China Program of the Chinese Academic of Sciences [2019]90 Guangxi Key Laboratory of Plant Conservation and Restoration Ecology in Karst Terrain 22-035-26 This work is supported by the National Natural Science Foundation of China (32460312), the Basic research fund of Guangxi Academy of Sciences (CQZ-D-1904), the Fundamental Research Fund of the Guangxi Institute of Botany, China (Guizhiye, 24010), the Key R&D Program of Guangxi, China (Guike AB22035038), the Light of West China Program of the Chinese Academic of Sciences ([2019]90), and the Guangxi Key Laboratory of Plant Conservation and Restoration Ecology in Karst Terrain (No. 22-035-26). The funders had no role in study design, data collection and analysis, decision to publish, or preparation of the manuscript.

==============================
With the expansion of the mining industry, environmental pollution from microelements (MP) and red mud (RM) has become a pressing issue. While bioremediation offers a cost-effective and sustainable solution, plant growth in these polluted environments remains difficult. Arundo donax is one of the few plants capable of surviving in RM-affected soils. To identify endophytic fungi that support A. donax in different contaminated environments and to inform future research combining mycorrhizal techniques with hyperaccumulator plants, we conducted a field experiment. The study compared endophytic fungal communities in A. donax grown in uncontaminated, MP soils contaminated with cadmium (Cd), arsenic (As), and lead (Pb), and RM-contaminated soils. Our findings showed that soil nutrient profiles differed by contamination type, with Cd concentrations in MP soils exceeding national pollution standards (GB 15168-2018) and RM soils characterized by high aluminum (Al), iron (Fe), and alkalinity. There were significant differences in the endophytic fungal community structures across the three soil types (p < 0.001). Co-occurrence network analysis revealed that endophytic fungi in MP soils exhibited competitive niche dynamics, whereas fungi in RM soils tended to share niches. Notably, Pleosporales sp., which accounted for 18% of the relative abundance in RM soils, was identified as a dominant and beneficial endophyte, making it a promising candidate for future bioremediation efforts. This study provides valuable insights into the role of endophytic fungi in phytoremediation and highlights their potential as resources for improving plant-microbe interactions in contaminated environments.

Introduction

Soil pollution resulting from human mining activities poses a significant environmental challenge. Mining operations increase the concentration of microelements in soils, leading to soil acidification or alkalization, which can be detrimental to plants, animals, and microorganisms (Rossi et al., 2017; Paez-Osuna et al., 2024). One of the most severe consequences of soil microelement pollution is its impact on ecosystem processes, notably by hindering biological degradation due to altered microbial biodiversity and soil properties (Hontoria et al., 2019). Alarmingly, this pollution does not diminish over time; for instance, in Sidi Village (Guangxi, China) elevated concentrations of cadmium (Cd), copper (Cu), lead (Pb), and zinc (Zn) persist in agricultural soils near an abandoned Pb-Zn mine, even after 40 years (Cao et al., 2018).

Additionally, waste from the mining industry, such as red mud—an industrial solid waste generated during aluminum (Al) extraction—exacerbates environmental pollution. Approximately 1.0 to 1.8 tons of red mud are produced for every ton of alumina extracted, depending on ore grade and production methods (Oprčkal et al., 2020). The increasing accumulation of red mud, coupled with its severe environmental impact, necessitates effective remediation strategies. Current research primarily focuses on resource recovery, such as utilizing red mud for ceramics and iron production (Liao, Zeng & Shih, 2015; Xiao et al., 2022). However, these methods often lack appropriate technology, incur high costs, and fail to fully address red mud pollution. Thus, there is an urgent need for eco-friendly and cost-effective remediation methods.

Bioremediation relies primarily on the synergistic effects of plants and microorganisms. Our research group has previously identified several hyperaccumulator plants, including Sedum, and Solanum nigrum, with Arundo donax (A. donax) emerging as a highly adaptable and resilient hyperaccumulator capable of suviving even in red mud, albeit with low biomass (Pu et al., 2020). To enhance the phytoremediation potential of A. donax in environments contaminated by trace elements and red mud, we aim to leverage mycorrhizal technology. By establishing symbiotic relationships between microorganisms and plants, we can effectively improve the host plant’s tolerance and accumulation abilities (Tedersoo, Bahram & Zobel, 2020; Rich et al., 2021). Thus, this study investigates the endophytic fungi associated with A. donax in polluted environments as a foundation for mycorrhizal technology.

The diversity and composition of plant endophytic fungi are influenced by various environmental factors, including pollution types, plant species, and soil characteristics (Chen et al., 2023). Endomycorrhizal fungi are often considered to be key factors in the bioavailability of metals to plants since they play a vital role in the activation and solidification of metals in soil (Liu et al., 2019). Therefore, it is essential to investigate the composition of endophytic fungi in relation to different types of pollution and soil properties.

This study aimed to identify the ecotypes of endomycorrhizal fungi of A. donax under unpolluted, trace element-contaminated, and red mud-contaminated conditions to develop effective phytoremediation strategies. Specifically, we conducted a field experiment focused on A. donax to (1) assess the diversity of endomycorrhizal fungi in its roots; (2) investigate the impact of different soil pollution types on the structure of endomycorrhizal fungi communities, and (3) analyze the changes in endophytic fungi diversity and the environmental factors driving these changes. This field-based approach provides valuable insights into the ecological interactions at play and highlights the potential of A. donax for phytoremediation in polluted environments.

Materials and Methods

Experimental design

The experiment was conducted at Guangxi Institute of Botany (25°01′N, 110°17′E, Guilin, China). A. donax stems containing auxiliary buds were selected as explants and propagated in vitro following the methodology of Xian et al. (2018). In vitro culture and seedling development were carried out under controlled laboratory conditions at the institute. The A. donax was cultivated ex situ until they reached uniform growth and a weight of about 75 g before transplantation.

Following 5 months of growth, uniform seedlings with well-developed root systems were selected for transplantation into three distinct soil types for in-situ remediation experiments. The first group was transplanted into unpolluted soil at the institute (UP). The second group was planted in agricultural soil polluted with microelements, specifically Cd, arsenic (As), and Pb, collected from Sidi Village, Yangshuo (MP). The third group was transferred to red mud (RM) sourced from the Pingguo County.

Sampling collection

After a 1-year cultivation period, soil samples were collected using a five-point composite sampling method and immediately stored at 4 °C to preserve their physical and chemical integrity for subsequent analysis. Root samples of A. donax were randomly collected and subjected to surface sterilization by immersing in 70% ethanol for 30 s, followed by 3% sodium hypochlorite for 5 min. The roots were then thoroughly rinsed with distilled water to remove residual disinfectants. The sterilized root samples were placed in a foam-insulated container with dry ice and transported via Shunfeng cold chain logistics. The samples arrived the next day at Shanghai Meiji Biomedical Technology Co., Ltd., where they were processed for endophytic fungal diversity analysis.

Environmental factor characteristics

Soil pH was determined using a pH meter (soil:water = 1:2.5, w/v) (ST3100; Ohaus Instruments, Parsippany, NJ, USA). Soil organic carbon (SOC) was determined using the potassium dichromate method, with results expressed as the percentage of SOC per gram of soil (Leifeld, Gubler & Gruenig, 2011). Total nitrogen (TN) and total phosphorus (TP) were measured using a UV spectrophotometer (Clever Chem 380; De Chem Tech, Hamburg, Germany). TN was quantified via the sodium salicylate method at 697 nm, with results reported as grams of TN per kilogram of soil. TP was determined using molybdenum-antimony spectrophotometry at 880 nm, expressed as grams of TP per kilogram of soil (Abrams, Metcalf & Hojjatie, 2014).

For microelement analysis, 0.2 g of air-dried soil was digested using a mixture of concentrated HNO3, HCl, and HF (4, 2, 2 ml). The digestion was performed in a microwave system at 120 °C for 5 min, followed by 200 °C for 20 min. The resulting solution was analyzed using inductively coupled plasma mass spectrometry (ICP-MS, Nexion 350; Perkin Elmer Instruments, Waltham, MA, USA) for trace elements including Pb, As, Cd, iron (Fe), Al, Titanium (Ti), magnesium (Mg), and calcium (Ca) (Lambkin & Alloway, 2000). Sodium (Na), potassium (K) concentrations were determined using a flame photometer (FP6430/FP6431/FP6432; Shanghai Jingke Industry Co., Shanghai, China).

High-throughput sequencing analysis of endophytic fungi

Plant samples were subjected to high-throughput sequencing by Shanghai Meiji Biomedical Technology Co., Ltd. The primers of ITS1F (5-CTTGGTCATTTAGAGGAAGTAA-3′) and ITS2R (5-GCTGCGTTCTTCATCGATGC-3′) were used for fungi (Lemons, Barnes & Green, 2017). The amplification program was 95 °C pre-denaturation for 3 min, 27 cycles (denaturation at 95 °C for 30 s, annealing at 55 °C for 30 s, and extension at 72 °C for 30 s), and finally extension at 72 °C for 10 min. The amplification system was 20 μL, 4 μL 5× FastPfu buffer, 2 μL dNTPs (2.5 mmol·L−1), 0.8 μL primer (5 μmol·L−1), 0.4 μL FastPfu polymerase, 10 ng DNA template, and use 2% agarose gel to recover the PCR product. AxyPrep DNA Gel ExtractionKit (Axygen-Biosciences, Union City, CA, USA) was used for purification, Tris-HCl elution, and 2% agarose electrophoresis detection. Quantification was performed using QuantiFluorTMST (Promega, Madison, WI, USA). The purified amplified fragments were used to construct a PE (2 × 250) library according to the standard operating procedures of the IlluminaMiSeq platform (Illumina, San Diego, CA, USA). High-throughput sequencing data are deposited in the National Center for Biotechnology Information (NCBI) (SRA: PRJNA1126329).

FLASH software was used to screen and optimize the original data sequence, non-repeating sequences were extracted from the optimized sequence and then the OTU clustering was performed based on 97% similarity (excluding single sequences), thus the effective sequences (similarity > 97%) were obtained. RDPclassifier (https://sourceforge.net/projects/rdp-classifier/) was used to annotate species classification for each sequence, compared it to the Silva database (SSU128) with a comparison threshold of 70%, and the mothur was used to calculate α diversity under different random sampling diversity index.

Statistical analysis

The experimental data were analyzed using SPSS 22.0 software. The Kolmogorov-Smirnov test was used to test the normality of data, and it was also checked for homoscedasticity of the residuals and autocorrelation via Durbin-Watson. One-way ANOVA was used to compare the difference among treatments. Tukey’s test was used to determine the significance at the level of p < 0.05. The data were presented as the mean ± standard error (SE) (n ≥ 3). The clustering status of the fungal community structure in the roots of A. donax in the UP, MP, and RM soils was analyzed using multidimensional scaling (NMDS) and similarity analysis (ANOSIM). Canonical correlation analysis (CCA) was used to reveal the correlation between microbial communities and environmental factors.

Linear discriminant analysis effect size (LEfSe) analysis was performed to evaluate the fungal compositions and determine the differentially abundant taxa at each taxonomic level for the root of A. donax associated with different soil types. A bipartite association network was used to visualize the associations between OTUs and the different polluted soil types.

Results

Soil properties

Compared to the unpolluted soil, the SOC and TN contents were significantly decreased in the MP and RM soils, the soil K and TP were significantly increased in the MP soils, the soil pH and Na contents were increased in the RM soils (Table 1).

Table 1 The soil properties of the unpolluted soil (UP), microelement polluted soil (MP), and red mud (RM).

Soil type	pH	Na (%)	K (%)	SOC (%)	TN (g/kg)	TP (g/kg)	
UP	6.82 ± 0.32b	1.9 ± 0.0b	0.33 ± 0.20 b	1.56 ± 0.09a	8.26 ± 0.01a	0.45 ± 0.01b	
MP	7.07 ± 0.58b	7.4 ± 0.2a	0.40 ± 0.10 b	0.92 ± 0.24b	6.06 ± 0.02b	0.59 ± 0.02a	
RM	10.15 ± 0.02a	1.0 ± 0.2c	20.27 ± 1.40a	0.34 ± 0.07c	3.97 ± 0.01c	0.47 ± 0.02b	
Note:

Data are presented as mean ± standard error (SE) (n ≥ 3). Significant differences between treatments were determined using one-way ANOVA followed by Tukey’s test. Different lower alphabetical letters indicate significant differences at p < 0.05.

The contents of Cd and Pb were highest in the MP soil and lowest in the UP soil, and their contents were 8.5 and 27 times higher in the MP soils than those in the UP soil (Table 2). The contents of Al and Fe in the RM soil were the highest, which were 2.1 and 3.2 times higher than those in UP soil, respectively (Table 2).

Table 2 The contents of trace elements of the unpolluted soil (UP), microelement polluted soil (MP), and red mud (RM).

Soil type	Al (g/kg)	Fe (g/kg)	As (mg/kg)	Cd (mg/kg)	Pb (mg/kg)	
UP	82.49 ± 0.26b	94.87 ± 2.29b	2.21 ± 0.53b	0.87 ± 0.00c	0.37 ± 0.02c	
MP	53.37 ± 2.45c	35.47 ± 0.14c	7.91 ± 0.19a	7.40 ± 0.04a	10.23 ± 0.70a	
RM	175.29 ± 4.66a	300.17 ± 14.31a	0.28 ± 0.01c	4.10 ± 0.15b	1.33 ± 0.04b	
Note:

Data are presented as mean ± standard error (SE) (n ≥ 3). Significant differences between treatments were determined using one-way ANOVA followed by Tukey’s test. Different lower alphabetical letters indicate significant differences at p < 0.05.

The α and β diversity of endophytic fungal community

There were 598,339 sequencing and 248 unique OTUs in the root samples of A. donax planted different soils. The fungal α diversity indexes such as Sobs, Shannon, and ACE were lower in the MP and RM soils than in the UP soil. The fungal Simpson diversity was significantly higher in the MP soils than in the UP and RM (Table 3).

Table 3 Summary of the Illumina MiSeq sequenced soil fungal richness and diversity indices for each soil type.

Soil type	Sobs	Shannon	Simpson	ACE	Chao1	
UP	69.00 ± 2.83a	2.25 ± 0.37a	0.20 ± 0.03b	75.30 ± 5.40a	91.75 ± 29.35a	
MP	39.50 ± 10.61b	0.59 ± 0.23b	0.79 ± 0.09a	43.68 ± 12.63ab	42.00 ± 12.02a	
RM	27.00 ± 9.90b	0.30 ± 0.09b	0.30 ± 0.09b	35.89 ± 11.55b	35.11 ± 8.17a	
Note:

Data are presented as mean ± standard error (SE) (n ≥ 3). Significant differences between treatments were determined using one-way ANOVA followed by Tukey’s test. Different lower alphabetical letters indicate significant differences at p < 0.05.

Multidimensional scaling analyzes showed that the structure of the fungal community in the roots of A. donax was significantly clustered in the UP, MP, and RM soils, and was different from each other soil types. Besides, the distribution areas of fungal communities were biggest in the MP soil, but smallest in the UP soil (Fig. 1).

Figure 1 Multidimensional scaling (NMDS) ordination of endophytic fungal communities from the root of A. donax among the different soil types (UP, MP, and RM).

Community composition of endophytic fungi

When the giant reed was transplanted from the UP soil to contaminated soil (MP and RM soils), the species and abundance of endophytic fungi colonizing its roots changed significantly (Fig. 2). The fungal community consisted of four different phyla while its relative abundance varied with soil types (Figs. 2B–2D). In the root samples of A. donax planted unpolluted soil, the first two dominant phyla were Basidiomycota and Ascomycota, and their relative abundances were 71.53% and 28.29%, respectively. When A. donax roots were transplanted to the MP and RM soils, the first two dominant phyla were Ascomycota (67.41%) and Basidiomycota (28.86%), and Ascomycota: (92.13%) and unclassified_k__Fungi (7.64%). Specifically, in the roots planted in the UP soil, the top three endophytic fungal species with the highest abundance were Marasmiellus tricolor (34%), Marasmiellus paspali (29%) and Pleosporales sp. (18%). In the MP and RM soils were Marasmiellus paspali (19%), Fusarium sp. (13%), Phialemoniopsis cornearis (13%), and Unclassified (Sordariomycetes) (65%), Unclassified (Hypocreales) (13%), Unclassified (Fungi) (7%), respectively.

Figure 2 Relative abundance (%) of the endophytic fungi in roots at the species (A) and phylum (B–D) levels.

LEfSe results showed that each soil types own unique fungal indicator taxa, from phylum (UP and MP soil) and Order (RM) to OTU levels (Fig. 3). Specifically, in the roots of A. donax in the UP soil, Basidiomycota, Agaricomycetes, and Marasmiaceae were enriched. Zygomycota, Thelephoraceae, Mortierella, and Sordariomycetes, Myrothecium cinctum were enriched in the roots planted in the MP and RM soils, respectively (Fig. 3).

Figure 3 Linear discriminant analysis effect size (LEfSe) cladogram of endophytic fungal communities from the root of A. donax among the different soil types.

Cluster and co-occurrence networks

In total, 81.9% OTUs were most strongly associated with only one soil type (Fig. 4, clusters 1–3). The 14.6% of OTUs were associated with two soil types (clusters 4, 5), while only 3.5% were associated with the three soil types (clusters 6). Fifteen OTUs in the cluster 4 were related to soil trace elements. Fourteen OTUs in the cluster 5 may be resistant to the MP and RM soils. Seven OTUs in the cluster 6 were necessary to the growth of A. donax and may not change with soil pollution (Fig. 4).

Figure 4 Bipartite association network between the soil types and the 196 associated OTUs. Node size represents the relative abundance of OTUs.

Co-occurrence network analysis showed that the UP and MP soils resulted in similar clustering patterns of the endophytic fungi, whereas the RM soil resulted in a different clustering pattern (Fig. 5). Compared with the UP soil, the negative correlations among OTUs were increased in the MP soil, meanwhile the positive correlations were increased in the RM soil.

Figure 5 Co-occurring network of endophytic fungi of A. donax at genus level in the unpolluted soil (A), microelement polluted soil (B) and red mud soil (C).

Sizes and colors of the nodes represent the relative abundance of the endophytic fungi. Solid lines with green and red colors indicate the negative and positive correlations, respectively. The width of lines reflects the strength of the correlation.

Fungal functions

A higher proportion of functional group fungi was observed in all soil types, especially the undefined saprotrophs (Fig. 6). Specifically, the relative abundance of the function undefined saprotrophs was highest in roots planted in the UP soil but lowest in the RM soil. Epiphyte-plant pathogen was found in both the MP and RM soils but not in UP soil, the average abundance of this pathogen was relatively low (Fig. 6).

Figure 6 Heatmap (A) and canonical correspondence analysis (CCA) (B) ordination plot based on the relationship between the environmental parameters and the community structure of ITs gene OTUs.

Driving factors of fungal communities

The correlation analysis revealed significant positive relationships among nutrient parameters (C, N, K, and P) as well as between Cd and pH. while, nutrient nutrient parameters were negatively correlated with Cd (Fig. 7A). Furthermore, Ascomycota (OTU7, OTU9, OTU10, etc.) exhibited negative correlations with nutrient parameters but positive correlations with Cd and pH. Conversely, Basidiomycota (OTU237, OTU202, OTU245, etc.) and certain Ascomycota (OTU201, OTU196, OTU191, etc.) were positively correlated with nutrient parameters and negatively correlated with trace elements (Cd and Pb) and pH. Moreover, these dominant species were widely present in UP, MP and RM treatments, showing significant positive correlations with pH and P, despite differences in composition among treatments (Fig. 7B).

Figure 7 Heatmap (A) and canonical correspondence analysis (CCA) (B) ordination plot based on the relationship between the environmental parameters and the community structure of ITs gene OTUs.

Discussion

Composition of endophytic fungal community in the roots of A. donax

The endophytic fungal community of A. donax is primarily composed of Basidiomycota, Ascomycota, unclassified_k__Fungi, and Zygomycota. Similarly, Dang et al. (2021) found that the endophytic fungi in the main root of Glycyrrhiza uralensis were predominantly Ascomycota and Basidiomycota. However, the richness and composition of these fungi vary significantly across different contaminated environments, with specific indicator taxa identified in each soil type. For instance, Basidiomycota dominated in UP treatment, while Ascomycota were more prevalent in MP and RM treatments. The correlation analysis suggests that increases in Cd and pH, or decreases in nutrient levels (C, N, P, and K), may drive a reduction in the abundance and diversity of Basidiomycota and an increase in Ascomycota. Moreover, the dominant endophytic species in A. donax across different treatments were primarily pathogenic fungi, such as Marasmiellus (a plant pathogen), Fusarium sp. (associated with citrus gummosis), and Phialemoniopsis (linked to various human infections) (Almaliky et al., 2013; Ito et al., 2017; Zakaria, 2023). One notable exception is Pleosporales sp., which has been reported to produce azaphilone compounds that show moderate inhibitory activity against three agricultural pathogens: Thielaviopsis paradoxa, Pestalotia calabae, and Glorosprium musarum (Cao et al., 2016). Thus, Pleosporales sp. may serve as a candidate strain for future bioremediation studies involving A. donax.

Interestingly, the three most abundant endophytic fungi colonizing the roots of A. donax in RM soil are unknown species. This could be attributed to both the extreme environment and potential technical limitations. RM soil represents an extreme environment with high potassium, salinity, alkalinity, and heavy metal concentrations. Microorganisms may undergo genetic mutation over long period of adaptation and evolution, leading to the emergence of unknown species better suited to such conditions. Previous studies have shown that fungal communities can adapt to extreme environments through traits like asexual reproduction, melanin-like pigment production, and flexible morphology (Gostincar et al., 2010). Additionally, Wang et al. (2023a) noted that the presence of unknown species in the identification of arbuscular mycorrhizal fungi in the rhizosphere of artificially planted pine forests might result from technological limitations in genetic sequencing. Future research should focus on identifying the functions of these unknown species to explore the potential applications of endophytic fungi that can tolerate extreme pollution.

Different soil types affect the diversity of endophytic fungal communities in the roots of A. donax

The composition and abundance of endophytic fungi in A. donax roots vary across different soil types, likely due to site-specific effects. Microbial communities in the rhizosphere are influenced by contaminants such as Pb, Cd, Zn, Mn, Fe, and S (Pfendler et al., 2024). Plants selectively filter specific microbial taxa from the soil, enriching them in the root zone (Cui et al., 2023). Stressful environments, such as contaminated soils, generally reduce the abundance and diversity of root endophytic fungi, while the abundance of pathogenic fungi tends to increase with greater levels of contamination (Wiewióra & Zurek, 2021; Cui et al., 2023). This may be due to the toxic effects of trace elements like Cd, As, and Pb, as well as high salinity and alkalinity, which decrease the abundance and diversity of heavy metal-sensitive endophytic fungi. Additionally, the negative impacts of pollution on plant growth hinder fungal colonization in the roots, with the severity of this limitation increasing with higher soil contamination levels (Becerra et al., 2023). Our study also observed that A. donax roots in MP and RM soils were enriched with plant pathogenic fungi, potentially altering the interactions among microorganisms across different sites.

Microelements and red mud contamination reduced the stability of the root endophytic fungal network of A. donax. The interactions among microorganisms influence the ecological niches and competitive dynamics of endophytic fungi (Huang et al., 2023; Zhang et al., 2024). Microelement pollution shifted many positive correlations among root endophytic fungi to negative, while the root endophytic fungi in the RM soil were closely and positively associated with each other. This may be due to the Cd, As, and Pb in microelement pollution will intensify the competition among root endophytic fungi on space and resource, and generally the diversity of root endophytic fungi decreasing with an increasing microelement concentrations (Li et al., 2016). However, except with high heavy metals, RM soil is also an extreme environment with high K and alkalinity (Ren et al., 2018). Plants and associated endophytic fungi in this environment may undergone adaptive evolution to cope with these specific stressors (Xie et al., 2024; Zeng et al., 2024). In summary, during the phytoremediation process, A. donax adjusts not only the composition of root endophytic fungi but also the relationships among microorganisms to adapt to varying stress conditions.

Environmental factors regulate the diversity and function of endophytic fungi in A. donax roots

The Cd content in MP contaminated soil exceeded the standard content of Class II soil (general land, GB 15168-2018) by about 12 times, and the K and P contents were also quite high. This may be due to excessive leaching of heavy metals from nearby mines or improper use of inorganic phosphate fertilizers over many years (Tian et al., 2023). According to the survey results of the National Bureau of Statistics (https://data.stats.gov.cn/), the pure volume of agricultural phosphate fertilizer application in Guangxi Zhuang Autonomous Region is 267,600 tons in 2022. The production process of phosphate fertilizer may cause heavy metal residues, leaving excess heavy metals and phosphorus in the soil with microelements (Lei et al., 2023). Moreover, this study found that arbuscular mycorrhizal fungi are not the main root endosymbiotic fungi in the three soil types. Studies have shown that some functions of arbuscular mycorrhizal fungi may be similar to the recruiting phosphorus-solubilizing bacteria and heavy metal-tolerant bacteria (Gutjahr et al., 2015). Therefore, in this study soil phosphorus may be not effectively utilized and remains.

In the MP soil, soil total nitrogen rather than other environmental factors has the biggest impact on the endophytic fungal community of A. donax roots. The function undefined saprotrophs has a high proportion in all soil types, and its relative abundance decreased following the order of the UP, MP, and RM soil. Saprophytic fungi is indispensable in the decomposition of soil organic matter, and the fungi is beneficial to soil nutrient recycling, plant growth, and ecosystem balance (Wang et al., 2021). Saprophytic fungi may also improve the plant’s ability to absorb water and mineral nutrients by forming a symbiotic relationship with plants (Cruz et al., 2022). Therefore, the reduction in the proportion of undefined saprotrophs may detrimentally affect soil nutrient cycling, which may be the main reason for the low organic carbon and nitrogen contents in the MP and RM soils.

Meanwhile, in the RM soil, the soil pH rather than other environmental factors has the biggest impact on the diversity of endophytic fungi in A. donax roots. Generally, the RM pollution is formed from sedimentation and stratification in areas with rich iron contents. During this process, abundant elements such as aluminum, iron, and potassium are accumulated, and the accumulation of soil potassium may contribute to a high-level soil pH in the RM soil (Wang et al., 2023b). In addition, soil nutrient contents (C, N, K, P) was negatively correlated with soil pH and Cd. The Basidiomycota was mainly positively correlated with C, N, K, and P, but negatively correlated with Cd, Pb, and soil pH. Previous studies found the Basidiomycota fungal populations related to macromolecule degradation are beneficial to inhibiting soil Cd activity (Zhao et al., 2023). Besides, the correlation between Ascomycota and environmental factors is bidirectional. In polluted environments, Ascomycota may be the most abundant fungal group (Cai et al., 2024). Although the Ascomycota species are diverse, generally these species ultimately contribute to the collaboration of microorganisms and plants, and thus promote the improvement of cadmium/chromium removal efficiency by shifting the proportion of dominant microbial-resistant taxa (Zhang et al., 2023). Overall, A. donax may assemble root endophytes according to soil types to deal with different soil pollutants.

Conclusions

The endophytic fungal community in A. donax roots is predominantly composed of Basidiomycetes, Ascomycetes, unclassified fungi, and Zygomycetes, with varying abundance and diversity across different contaminated soil types. Notably, higher concentrations of trace elements and elevated pH levels, coupled with lower nutrient availability in trace element and red mud-contaminated soils, tend to suppress Basidiomycete populations while promoting Ascomycete proliferation. Furthermore, A. donax may adapt to extreme environments, such as red mud, by colonizing novel (unknown) species and facilitating interactions among endophytic fungi. Remarkably, Pleosporales sp. emerges as a highly promising endophytic fungus for remediation in contaminated soils. Future research should focus on identifying the functional roles of dominant unknown species and evaluating the remediation potential of Pleosporales sp. to investigate the synergistic effects of endophytic fungi and hyperaccumulator A. donax in the remediation of trace element and red mud contamination.

Supplemental Information

Supplemental Information 1 Raw data on physical and chemical properties of different types of soils.

Supplemental Information 2 RM Processing Group Microbial High Throughput Information Data 1.

Supplemental Information 3 RM Processing Group Microbial High Throughput Information Data 2.

Supplemental Information 4 RM Processing Group Microbial High Throughput Information Data 3.

Supplemental Information 5 RM Processing Group Microbial High Throughput Information Data 4.

Supplemental Information 6 RM Processing Group Microbial High Throughput Information Data 5.

Supplemental Information 7 RM Processing Group Microbial High Throughput Information Data 6.

Supplemental Information 8 MP Processing Group Microbial High Throughput Information Data 1.

Supplemental Information 9 MP Processing Group Microbial High Throughput Information Data 2.

Supplemental Information 10 MP Processing Group Microbial High Throughput Information Data 3.

Supplemental Information 11 MP Processing Group Microbial High Throughput Information Data 4.

Supplemental Information 12 MP Processing Group Microbial High Throughput Information Data 5.

Supplemental Information 13 MP Processing Group Microbial High Throughput Information Data 6.

Supplemental Information 14 UP Processing Group Microbial High Throughput Information Data 1.

Supplemental Information 15 UP Processing Group Microbial High Throughput Information Data 2.

Supplemental Information 16 UP Processing Group Microbial High Throughput Information Data 3.

Supplemental Information 17 UP Processing Group Microbial High Throughput Information Data 4.

Supplemental Information 18 UP Processing Group Microbial High Throughput Information Data 5.

Supplemental Information 19 UP Processing Group Microbial High Throughput Information Data 6.

Additional Information and Declarations

Competing Interests

Author Contributions

DNA Deposition

Data Availability

The authors declare that they have no competing interests.

Xiaohui Wang conceived and designed the experiments, performed the experiments, analyzed the data, prepared figures and/or tables, authored or reviewed drafts of the article, and approved the final draft.

Yao Wang performed the experiments, analyzed the data, prepared figures and/or tables, and approved the final draft.

Yingqiang Sun analyzed the data, prepared figures and/or tables, and approved the final draft.

Keyi Wang performed the experiments, prepared figures and/or tables, and approved the final draft.

Junbo Yang analyzed the data, authored or reviewed drafts of the article, and approved the final draft.

Danjuan Zeng performed the experiments, authored or reviewed drafts of the article, and approved the final draft.

Ling Mo performed the experiments, authored or reviewed drafts of the article, and approved the final draft.

Jianxiong Liao conceived and designed the experiments, authored or reviewed drafts of the article, and approved the final draft.

Qianshu Peng analyzed the data, prepared figures and/or tables, and approved the final draft.

Yu Yao analyzed the data, prepared figures and/or tables, and approved the final draft.

Gaozhong Pu conceived and designed the experiments, prepared figures and/or tables, authored or reviewed drafts of the article, and approved the final draft.

The following information was supplied regarding the deposition of DNA sequences:

The high-throughput sequencing data are available at the National Center for Biotechnology Information (NCBI) SRA: PRJNA1126329.

The following information was supplied regarding data availability:

The raw measurements are available in the Supplemental File.

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
