# Peer review of "Soil polluted system shapes endophytic fungi communities associated with Arundo donax: a field experiment"

_PeerJ, doi:10.7717/peerj.18789_

## Round 0.1 · original submission · Major Revisions

We have concluded the revision of your manuscript. Three expert reviewers have evaluated your manuscript and their comments can be seen below. Due to the extent of their comments and concerns, I have made the decision that a major revision is required. as you prepare your manuscript for resubmission, please ensure that you include a careful accounting of every suggestion and how it was incorporated or otherwise addressed in the manuscript so that the reviewers and I have the easiest time verifying the improvements and clarifications.

Reviewer 1 ·

Basic reporting

The work is simple, scientific content of this manuscript weak

Experimental design

The experimental design is poor, author needs to perform additional studies. the present form preliminary level only

Validity of the findings

Additional studies required and obtained results should be presented in depth and the author should improve the discussion section by adding more comparisons with published literature

Additional comments

The concept of the work is interesting, however, I am unable to recommend this manuscript in this present form for publication since the work is preliminary level and additional studies are needed to strengthen the manuscript as scientific merit

Reviewer 2 ·

Basic reporting

• The language used sometimes lacks clarity. I suspect that the authors used AI, which may have caused some sentences to lose their meaning.
• The title is OK.
• The abstract needs to be re-written as the authors made a general summary without giving specific details about the results. It's important for them to share the findings using percentages. Also, they should clearly explain the importance of the study in this part.
• In introduction the literature is relevant, but some of the references are too old. Please include more recent references in the list if possible, avoiding those that are over 10 years old.
• The methods section needs to be thoroughly explained to ensure reproducibility.
• The figures are relevant, but the labels are difficult to read, inconsistent, and the figures lack explanations in their legends.

Experimental design

• The statistical design and analysis are adequate, but some of the methods are not well explained.

Validity of the findings

• The work is novel, but it needs to be presented more effectively.
• The conclusion lacks clarity at certain points, which suggests that the authors may have used AI, leading to a misunderstanding of their intended message.

Additional comments

• Line 19: write Cd, As and Pb in full as you have written Iron (Fe) in line number 38.
• Line 21: what is meant by OTUs.
• The referencing style is inconsistent, with formats such as “(Gremion et al., 2004; Rossi et al., 2017)” and “(Hassan et al. 2011, Hontoria et al. 2019)”. I suggest using Endnote for consistent referencing.
• Line 45: What do you mean by “mud8[liaolaoshi]”?
• Line 43-44: Replace “Arundo donax (A. donax)” with “A. donax”
• Line 60-61: What do you mean by “ To this end, a field experiment was …..”?
• Line 80-81: For how long you soaked the roots in 70% ethanol?
• Line 81: What was the concentration of sodium hypoclorite? For how long you soaked the roots in it?
• Line 81-82: I haven't heard until now that distilled water can disinfect roots or anything else. Please correct the statement.
• Line 82-83: How you transported the disinfected roots to Shanghai Meiji Biomedical Technology Co., Ltd.
• Line 83: The “s” of Shanghai should be written in capital.
• Line 86-91: Please explain the methods.
• Line 169: I understand that scientific names should be written in italics, but why have you written the common words in italics?

The paper should be reviewed by a native speaker before submitting the revised version!

Reviewer 3 ·

Basic reporting

Clear in writing
Reference style is not uniform in the text
Tables are fine but the some figure is difficult understand

Experimental design

It is not clear about the experimental layout and area used for the study

Validity of the findings

Novelty of the study and impact/future recommendation is missing

Additional comments

Regarding the manuscript, the authors made very good piece of work on “Soil polluted system shapes endophytic fungi communities associated with Arundo donax: A field experiment”. The endophytic fungi communities studied with the plant Arundo donax for the possible phytoremediation in different soil types. The research looks very interesting and well set up of experiment, still it needs some clarification for further modification as follows;
1. How the pollute site is selected? What is the criteria for the selection of Arundo donax?
2. The field experiment layout or how much area is used with phytoremediation purpose with the endophytic fungi and what is the significance of the present study should be mentioned
3. What about the site characteristics with relation to microelement concentration?
4. What is the contamination level of each site with respects to standard or permissible limit?
5. The table 2 it is clear that the unpolluted site also having As concentration 2.21g/kg, I think this is very high concentration as per as arsenic pollution is concern? Please clarify.
6. With relation to above point, same is the case for RM soil with relation to Pb concentration
7. Table 3. Chao 1 diversity indices, was found 42 for MP soil, which is higher than RM soil, is the microelements contamination increase the diversity, which is better in this case. please clarify
8. Some of the figure like figure 6 also signifies max diversity under microelement contaminated soil.
9. The figure 7 is difficult to understand.
10. The SOC level of each soil is very less, does it classify as degraded soil with minimal carbon
11. Conclusion is not clear, only soil types is the key to abundance and diversity of endophytic fungi, or microelements concentration. What about the environmental factors for the endophytic fungi growth and climatic adaptability?
12. What is the recommendation out of this study? Which endophytic fungi is dominating with the plant used for this study?
13. The future research gaps also need to be addressed.

---

## Round 0.2 · Minor Revisions

I am sincerely sorry that it has taken so long to come to a decision on your manuscript. The prior reviewers were invited, but unfortunately no reviewers responded. Therefore, I have read through your letter to reviewers and the track changes and clean versions of your manuscript. I commend you on the thoroughness of your attention to the concerns of the reviewers that have markedly improved the manuscript and the figures. I found some minor errors that need to be corrected prior to accepting the manuscript - please see the attached PDF for details.

---

## Round 0.3 · accepted · Accept

Thank you for making the suggested minor changes. I am satisfied with them and am happy to accept this manuscript for publication.